# Non-Immersive Virtual Reality Telerehabilitation System Improves Postural Balance in People with Chronic Neurological Diseases

**DOI:** 10.3390/jcm12093178

**Published:** 2023-04-28

**Authors:** Michela Goffredo, Chiara Pagliari, Andrea Turolla, Cristina Tassorelli, Sonia Di Tella, Sara Federico, Sanaz Pournajaf, Johanna Jonsdottir, Roberto De Icco, Leonardo Pellicciari, Rocco Salvatore Calabrò, Francesca Baglio, Marco Franceschini

**Affiliations:** 1Neurorehabilitation Research Laboratory, Department of Neurological and Rehabilitation Sciences, IRCCS San Raffaele Roma, 00163 Rome, Italy; 2IRCCS Fondazione Don Carlo Gnocchi ONLUS, 20148 Milan, Italy; 3Department of Biomedical and Neuromotor Sciences (DIBINEM), Alma Mater University of Bologna, 40138 Bologna, Italy; 4Unit of Occupational Medicine, IRCCS Azienda Ospedaliero-Universitaria di Bologna, 40138 Bologna, Italy; 5Department of Brain and Behavioral Sciences, University of Pavia, 27100 Pavia, Italy; 6Headache Science & Neurorehabilitation Center, IRCCS Mondino Foundation, 27100 Pavia, Italy; 7Laboratory of Healthcare Innovation Technology, IRCCS San Camillo Hospital, 30126 Venice, Italy; 8IRCCS Istituto delle Scienze Neurologiche di Bologna, 40139 Bologna, Italy; 9IRCCS Bonino-Pulejo, 98124 Messina, Italy; 10Department of Human Sciences and Promotion of the Quality of Life, San Raffaele University, 00166 Rome, Italy

**Keywords:** telerehabilitation, neurorehabilitation, chronic neurological diseases, Parkinson disease, Multiple Sclerosis

## Abstract

Background: People with chronic neurological diseases, such as Parkinson’s Disease (PD) and Multiple Sclerosis (MS), often present postural disorders and a high risk of falling. When difficulties in achieving outpatient rehabilitation services occur, a solution to guarantee the continuity of care may be telerehabilitation. This study intends to expand the scope of our previously published research on the impact of telerehabilitation on quality of life in an MS sample, testing the impact of this type of intervention in a larger sample of neurological patients also including PD individuals on postural balance. Methods: We included 60 participants with MS and 72 with PD. All enrolled subjects were randomized into two groups: 65 in the intervention group and 67 in the control group. Both treatments lasted 30–40 sessions (5 days/week, 6–8 weeks). Motor, cognitive, and participation outcomes were registered before and after the treatments. Results: All participants improved the outcomes at the end of the treatments. The study’s primary outcome (Mini-BESTest) registered a greater significant improvement in the telerehabilitation group than in the control group. Conclusions: Our results demonstrated that non-immersive virtual reality telerehabilitation is well tolerated and positively affects static and dynamic balance and gait in people with PD and MS.

## 1. Introduction

Over the past 30 years, the burden of chronic neurological diseases is continuously augmenting due to population increase and aging, with dramatic medical and economic implications [1]. Chronic neurological diseases such as Parkinson’s Disease (PD) and Multiple Sclerosis (MS), have a significant long-term impact on Activities of Daily Living progressively reducing the Quality of Life (QoL) [2,3,4]. For these reasons, the management of PD and MS is a health priority of the WHO considering the increasing demand for treatment, rehabilitation, and support services [5]. 

Both PD and MS result from progressive neuronal dysfunction and neuronal cell death leading to progressive disability [6]. Patients with PD or MS suffer motor and non-motor signs and symptoms, including cognitive dysfunction [7,8], behavioral problems [9,10], motor and gait dysfunction [11,12], and loss of balance [13]. Although PD and MS are different in pathophysiology and clinical presentation [6], postural balance disorder leads to a high prevalence of falls in chronic neurological diseases regardless of diagnosis [14,15,16] and the implementation of a regular prolonged training program is crucial [17,18,19,20,21]. The recent literature analyzed the impact of exercise on balance and fall prevention in people with stroke, PD, and MS, showing that physical activity has a positive effect on increasing postural control [22,23], especially when it is associated with exergames and virtual [24,25]. In this context, the use of digital platforms for the continuous delivery of rehabilitation services represents the new frontier for the management of chronic neurological diseases [26,27]. This approach ensures synergy between different healthcare professionals for a multidimensional diagnostic and rehabilitation service and empowerment of the person in the management of their care plan.

Considering that people with chronic neurological diseases may have difficulty in achieving outpatient rehabilitation services because of economic, geographic, and social-distancing barriers [28], a solution to guarantee the continuity of care may be telerehabilitation, i.e., the delivery of clinical rehabilitation services for evaluation and treatment at distance [29]. The restrictions imposed by the COVID-19 pandemic emphasized, even more, the fundamental role of telerehabilitation in situations where access to rehabilitation facilities is limited or, in some cases, impossible [28,29,30,31].

The scientific literature on telerehabilitation shows that it is a feasible, easily accepted, and effective treatment in neurological patients [32,33,34] since it allows optimization of the timing, intensity, and personalization of the rehabilitative intervention. Moreover, telerehabilitation is particularly suitable for people with PD and MS who need repeated and prolonged rehabilitation cycles due to the chronic and progressive nature of the diseases [34,35,36,37,38,39,40]. The literature on telerehabilitation in people with PD is limited and controversial, as evidenced in recent systematic reviews [25,34,36,37,38]: it seems to be feasible and effective in improving only a subset of clinical and non-clinical aspects of PD (balance and gait, speech and voice, quality of life) similarly to the conventional treatments. Although systematic reviews on telerehabilitation in people with MS [39,40] reported a limited number of RCTs on this topic, telerehabilitation seems to be beneficial in functional activities, impairments, and participation. These outcomes have been confirmed by Truijen et al. [38], who found that platform and exergaming-based telerehabilitation improves balance in individuals with central neurological disorders, and our recent RCT [41] registered significant improvement in balance, postural control, and dynamic walking in the telerehabilitation group. Although the current findings are encouraging, even though limited, further studies on virtual reality and exergaming-based balance telerehabilitation for improving postural stability are needed [41]. This study intends to expand the scope of previously published research on the impact of telerehabilitation on quality of life in an MS sample [40], testing the impact of this type of intervention in a larger sample of neurological patients also including PD individuals on postural balance.

Therefore, in this study, we analyzed collectively the data obtained from people with PD and MS using motor outcome measures (the mini-Balance Evaluation Systems Test and the Timed Up-and-Go) collected for both clinical samples diseases with the purpose of investigating the efficacy of non-immersive virtual reality telerehabilitation system on the motor functioning of these two paradigmatic examples of chronic neurological diseases, compared to at-home conventional rehabilitation. We hypothesized positive effects of telerehabilitation in improving static and dynamic balance (primary outcome measure), gait, and cognitive functions, and in reducing the patient’s risk for falls. 

## 2. Materials and Methods

### 2.1. Participants

A sample of individuals diagnosed with PD (according to the United Kingdom PD Society Brain Bank criteria [42]) or MS (according to McDonald’s criteria [43]) was recruited between 2017 and 2020 within two multicenter randomized controlled trials that involved five Italian rehabilitation hospitals (Istituti di Ricovero e Cura a Carattere Scientifico—IRCCS) of the Italian Neuroscience and Rehabilitation Network (https://www.reteneuroscienze.it/en; accessed on 7 February 2023). The subjects were enrolled if they met the following eligibility (inclusion and exclusion) criteria:Between 25 and 70 years of age;Stage of disease: mild to moderate as documented by Hoehn and Yahr (H&Y score range between 2 and 3—PD group) or Expanded Disability Status Scale (EDSS score ≤ 6.5—MS group);Absence of cognitive impairment measured by the MoCA total score ≥ 18 [44] and sufficient cognitive and linguistic level to understand and comply with study procedures;Stabilized drug treatment for at least 3 months before starting this study;Absence of moderate and severe dyskinesia and freezing episodes as documented by MDS-UPDRS (PD group);No other neurologic conditions different from MS or PD;No psychiatric complications or personality disorders, as indicated in the medical documentation;Absence of severe primary sensory deficits such as blurring or low vision, severe hearing loss and speech disorder

Written and informed consent was obtained for all participants. None of the participants were involved in other experimental trials during the entire duration of the present study.

### 2.2. Rehabilitation Procedures

All enrolled subjects were randomized into two groups: (1) the Intervention Group (IG) which received a non-immersive virtual reality telerehabilitation system and (2) the Control Group (CG) which received at-home conventional rehabilitation. The randomization procedure was applied to people with PD and to MS separately.

### 2.3. Intervention Group (IG)

The IG underwent 30–40 sessions (5 days/week, 6–8 weeks; 45 min each session) of motor, and cognitive rehabilitation exercises using the VRRS Tablet home telerehabilitation system (Khymeia Srl, Noventa Padovana, Italy). The motor exercises were performed using inertial sensors for the acquisition and processing of the movement performed by the patient. These data were shown to the patient with visual and auditory feedback in a serious game environment. The exercises covered the rehabilitation of balance and lower limbs (e.g., maintaining balance on one leg, marching in place, standing on tiptoe, squatting, etc.). The therapists involved in the study defined the protocol of exercises in telerehabilitation mode customized according to the characteristics and needs of the patient.

### 2.4. Control Group (CG)

The CG underwent 30–40 sessions (5 days/week, 6–8 weeks; 45 min each session) of at-home treatments without the use of any technological devices. The CG rehabilitation was an active comparator treatment and consisted of a written home-based self-administered booklet with conventional motor and cognitive activities tailored for each patient. The motor exercises included the rehabilitation of balance and lower limbs (e.g., maintaining balance on one leg, marching in place, standing on tiptoe, squatting, etc.). The intensity and duration of the CG were the same as the IG.

### 2.5. Outcome Measures

Clinical assessment was performed at baseline (T1) and at the end of the treatment (T2) both in people with PD and MS. All investigators and outcome assessors were blinded to the type of treatment. The following outcome measures were administered by assessors blinded to the intervention groups:The mini-Balance Evaluation Systems Test (mini-BESTest) is a shortened version of the Balance Evaluation Systems Test. It is composed of a 14-item scale that evaluates balance with a total score of 28. Items are grouped into the following four subcomponents: anticipatory postural control (max score = 6), reactive postural control (max score = 6), somatosensory orientation (max score = 6), and dynamic walking (max score = 10). A summary of the subcomponents and the items of the mini-BESTest is depicted in Table 1. The mini-BESTest has been shown to have good psychometric properties in both PD and MS [45,46].The Timed Up-and-Go (TUG) test which involves rising from a seated position, walking to a pre-determined location, turning, and returning to a seated position, is a common test used to assess functional mobility, dynamic balance, and walking ability. The score is the time required to perform the following tasks: standing up from a chair; walking 3 m: turning around, walking back to the chair and sitting down. The validity and reliability of the TUG in people with PD and MS have been published [47,48]. TUG performance has been associated with mobility status and fall risk [49,50]. The TUG test used was the subtest included in the mini-BESTest.The Timed Up-and-Go-test Dual-task (TUG-D) is a dual-task measure of functional mobility that evaluates balance with a simultaneous cognitive task. The TUG-D score is the time required to perform the TUG when the following cognitive task is added: while walking, the participant counts backward in threes from a randomly chosen start number between 60 and 100 to avoid a learning effect. The TUG-D performance on the TUG-D represents a significant predictor of future falls in people with PD and MS [51,52]. The TUG-D test used was the subtest included in the mini-BESTest.The Montreal Cognitive Assessment (MoCA) is a rapid screening instrument for mild cognitive dysfunction. It assesses different cognitive domains: attention and concentration, executive functions, memory, language, visuo-constructional skills, conceptual thinking, calculations, and orientation. The total MoCA score is 30 points. The MoCA has been recognized as a valid and sensitive instrument to identify cognitive impairment in people with PD and MS [53,54].

Moreover, the following outcome measures were administered for people with PD at T1 and T2:The MDS-Unified Parkinson’s Disease Rating Scale (MDS-UPDRS) is a multimodal scale assessing impairment and disability consisting of four parts. Part I assessed non-motor experiences of daily living. Part II assessed motor experiences of daily living. Part III assessed the motor signs of PD. Part IV assessed motor fluctuations and dyskinesias. MDS-UPDRS Total Score equals the sum of Parts I, II, and III (Range 0–236). A higher score indicated more severe symptoms of PD [55].The Parkinson’s Disease Questionaire-8 (PDQ-8) is a short-form version of the Parkinson’s Disease Questionaire-39. It is a self-administered questionnaire, used to measure the quality of life in people with PD [56]. The total PDQ-8 score is 32 points.

The following outcome measures were administrated for people with MS at T1 and T2:
The Multiple Sclerosis Quality of Life 54 (MSQoL-54) is a structured, self-report questionnaire for measuring health-related quality of life in MS [57] consisting of four parts. The MSQoL-54 consists of 12 subscales and two single items. Each subscale is scored from 0 to 100, with higher scores indicating a better QoL. Subscale scores can be weighted and summed to generate Physical Health Composite Score (MSQOL-54_PHCS) and Mental Health Composite Score (MSQOL-54_MHCS).

The primary outcome of the study was the mini-BESTest total score.

### 2.6. Statistical Analysis

The database analyzed in this study was composed of people with MS from our previous study [38] to which people with PD have been added. A sample size of 60 patients (30 per arm) achieves 95% power to detect a difference of 2.0 (standard deviation = 2.0; Cohen’s d = 1) in the primary outcome [58] in a design with two repeated measurements according to the literature on the psychometric performance of the Mini-BESTest in patients with balance disorders [59], assuming an alpha error of 0.05, and 5% dropout rate of patients. A priori sample size was calculated using G*Power 3.1 software. Statistical analyses were carried out by using jamovi (Version 2.3) software (https://www.jamovi.org; accessed on 7 February 2023). Summary statistics are expressed as frequencies, percentages, means and Standard Deviations (SD), median and Interquartile Range (IQR). Comparisons between the two groups (IG and CG) of baseline clinical features in the full sample and people with PD and MS separately using parametric (independent samples *t*-tests, analysis of covariance—ANCOVA) or corresponding non-parametric (independent samples Mann–Whitney U test) tests, as appropriate. Variable distribution was inspected through histograms and skewness and kurtosis statistics were calculated to assess normality. When variables violated the assumption of normal distribution, the natural logarithmic [ln] transformation was applied. A modified intention-to-treat method was implemented [60]. Missing data for the outcome measures comprised in the principal dataset were handled according to a single imputation procedure replacing missing values with the median value in each group at a specific time point in case of missing data less than 5% [61]. Outcome measures were analyzed using the jamovi module General analyses for linear model [62] (retrieved from https://gamlj.github.io; accessed on on 7 February 2023). Generalized Linear Mixed Models (GLMMs) were performed to evaluate score differences across the two time points (T1 and T2). Different test scores were used as dependent variables (one for each model), and the effects (Time, Group, Pathology) were considered independent variables. Time, Group, Pathology, and their interaction (Time✻Group, Time✻Pathology, Group✻Pathology, Time✻Group✻Pathology) were included in each model as fixed effects. To account for subject specific variability, each subject was used as a random factor in all the models. GLMMs for the cognitive outcome (MoCa Test), also included age and education as covariates. The final models were the following: Motor Outcome measure ~1 + Time + Group + Pathology + Time:Group + Time:Pathology + Group:Pathology + Time:Group:Pathology + (1|Subject), Cognitive Outcome measure ~1 + Time + Group + Pathology + Age + Education + Time:Group + Time:Pathology + Group:Pathology + Time:Group:Pathology + (1|Subject). Effects sizes (partial eta-squared pη^2^) for the posthoc tests performed to interpret significant fixed effects were calculated by R Studio software (Version 1.4.1106), and the magnitude of effects was interpreted as follows: small (pη^2^ = 0.01), medium (pη^2^ = 0.06), and large (pη^2^ = 0.14) effects [63]. The statistical significance was set at *p*-value < 0.05 for all analyses.

## 3. Results

A sample of 150 participants met the inclusion criteria and were included in the study: 80 people with PD and 70 with SM. Of the sample, 75 participants were allocated to the IG (PD = 40; MS = 35) and 75 to the CG (PD = 40; MS = 35). A sample of 132 participants (IG = 65; CG = 67) completed the baseline (T0) and post-treatment (T1) evaluations, and 18 subjects dropped out (IG = 10; CG = 8). None of the drop-outs occurred during the study for treatment-related reasons, and no participant experienced any adverse event during treatment. Baseline demographics and clinical data in the full sample, people with PD, and people with MS are detailed in Table 2. No differences were registered between IG and CG at T0. The baseline demographics and clinical data in the IG and CG are detailed in the Appendix A. Results of the GLMMs were performed on each outcome measure to verify whether the two treatments had different effects on the common outcomes in the full sample of patients with chronic neurological diseases (Table 3). 

All clinical outcomes significantly improved between T0 and T1 in both groups except for Mini-BESTest Reactive postural control and Somatosensory orientation subcomponents. When considering the primary outcome (i.e., the Mini-BESTest total score), the IG showed a greater improvement than the CG (effect of interaction Time✻Group: *p*-value = 0.020; posthoc comparison: IG T0 vs. T1 *p*-value < 0.001, pη^2^ = 0.13) with an improvement of about 24% concerning the maximum score achievable. A significant effect of factor Time was observed for the Mini-BESTest Anticipatory postural control subcomponent (*p*-value < 0.001, pη^2^ = 0.10) showing an improvement in both groups. When considering the Mini-BESTest Dynamic walking subcomponent, the IG showed a significantly higher improvement (effect of interaction Time✻Group *p*-value = 0.011; posthoc comparison: IG T0 vs. T1, *p*-value < 0.001, pη^2^ = 0.13) than the CG (see Figure 1). A significant effect of factor Time was observed for the TUG[ln] (*p*-value = 0.002, pη^2^ = 0.07) showing an improvement in both IG and CG. In TUG-D[ln], the IG improved more than the CG (effect of interaction Time✻Group: *p*-value = 0.048; posthoc comparison: IG T0 vs. T1 *p*-value < 0.001, pη^2^ = 0.11). A general effect of Time emerged for the global cognitive functioning (MoCA: *p*-value = 0.003, pη^2^ = 0.07) with an improvement of about 19% with respect to the maximum score achievable in both groups.

The triple interaction (Time✻Group✻Pathology) analysis showed significant differences in the TUG-D[ln] only (effect of interaction Time✻Group✻Pathology *p*-value = 0.014). Specifically, the posthoc analysis revealed that the people with MS who underwent the telerehabilitation intervention improved the velocity in performing the TUG-D[ln] after treatment, whereas no amelioration was observed in the people with MS who underwent the CG intervention (posthoc comparison: people with MS IG T0 vs. T1, *p*-value < 0.001, pη^2^ = 0.09). No significant differences were found in the people with PD Figure 2 depicts the results on TUG-D[ln].

## 4. Discussion

Considering the importance of a regular sustained program for balance rehabilitation and fall prevention in chronic neurological diseases [19,20,21], this study moves from previously published findings on the impact of telerehabilitation on quality of life in an MS [40] expanding the research scope and the target sample. Hypothesizing to achieve positive effects of telerehabilitation in balance capacity, we tested the effect of such intervention on postural balance in a larger sample of neurologic patients, including not only patients with MS but also with PD.

An RCT cohort of 132 subjects with chronic neurological diseases was considered, representing we analyzed 87% of the fulfilled dataset (and the remaining 13% of the dataset was dismissed due to dropout). Our results showed that the non-immersive virtual reality telerehabilitation system implemented in this study was feasible and easily accepted in people with PD or MS, in line with previously published studies [34,35,36].

This study showed that the non-immersive virtual reality telerehabilitation system was effective and allowed to optimize the timing and intensity of the rehabilitation intervention. These findings are particularly relevant in the case of economic, geographic, and social-distancing barriers than may hinder people with chronic neurological diseases from achieving outpatient rehabilitation services [28]. In this context, the non-immersive virtual reality telerehabilitation system is a promising approach to ensure continuity of care [29].

The postural balance assessed with the Mini-BESTest total score registered a higher improvement in the IG compared to the CG: subjects who conducted non-immersive virtual reality telerehabilitation system amended postural control more than their peers who conducted at-home conventional rehabilitation without the use of any technological devices. The outcomes obtained agree with the literature on the effects of telerehabilitation in improving postural control in individuals with chronic neurological diseases [38], PD [39,64], and MS [24,34,40,42,65,66]. On the other hand, in people with PD, the superiority of telerehabilitation in increasing postural stability has not been confirmed in some studies [39,62]. Specifically, Gandolfi et al. [39] and Seidler et al. [62] demonstrated that telerehabilitation improved balance and gait functions similar to conventional treatments.

The impact of the non-immersive virtual reality telerehabilitation system on postural control had been further explored by analyzing the dynamic balance and gait by means of the Mini-BESTest subcomponents, TUG, and TUG-D. The results evidenced that both the TI and the CG improved the Mini-BESTest “anticipatory postural control” subcomponent and TUG at the end of the treatment, without registering any between-group differences. Compared with the CG, the TI performed significantly better at the Mini-BESTest “dynamic walking” subcomponent and TUG-D. Thus, subjects who conducted non-immersive virtual reality telerehabilitation registered a higher performance in the execution of the following motor tasks: change in gait speed; walk with head turns; walk with pivot turns; step over obstacles and timed up and go with a dual task. These results are in agreement with the literature that showed an improvement in the ability to control balance and change gait speed and direction in subjects who underwent telerehabilitation [38,62,67].

Similar outcomes were also obtained in studies on the application of virtual reality technology in rehabilitation, which positively improved movement velocity, balance, and gait [68,69]. The results on the ability to execute a dual task (i.e., the TUG-D) are representative of the potential for non-immersive virtual reality telerehabilitation systems to improve both motor and cognitive functions [52]. To this extent, Intzandt et al. [70] analyzed how different types of rehabilitation can influence motor function (gait) and cognition in people with PD, finding that goal-based training can mitigate both motor and non-motor symptoms (such as fatigue, depression, apathy, and cognitive impairment) which can also influence motor performance.

The importance of the obtained outcome in favor of the non-immersive virtual reality telerehabilitation is also enhanced by the fact that the difficulty in executing a motor task paired with a simultaneous cognitive task has been associated with falls in people with chronic neurological diseases [52,71,72]. Furthermore, dual-task activities involving both motor and cognitive resources constitute a significant portion of most activities of daily living, and thus improving these activities has a major impact on participation and quality of life [73,74,75].

The analysis of the effects of telerehabilitation on cognitive functions revealed that both treatments improved the MoCA total score, and no differences were found between groups. Indeed, this outcome was influenced by the criteria for subject recruitment which included only participants without severe cognitive impairment (MoCA total score ≥ 18). Although the primary goal of the telerehabilitation treatments was the rehabilitation of balance and lower limb functions, the obtained outcomes on cognitive performance are encouraging and consistent with the literature [76,77], calling for future studies on cognitive telerehabilitation in chronic neurological diseases.

The analysis of differences between PD and MS revealed that the positive improvement of telerehabilitation in postural balance (Mini-BESTest total score) and ability to change gait speed and direction (Mini-BESTest “dynamic walking” subcomponent) was registered in both pathologies. Thus, the hypothesis of the study, i.e., telerehabilitation is effective in improving balance in both people with PD and SM, is reinforced and justified. On the other hand, the ability to execute a dual-task (i.e., the TUG-D) registered a significant difference between pathologies: subjects with MS who conducted the non-immersive virtual reality telerehabilitation improved the TUG-D more than their peers with PD. There is supportive evidence in the literature for the use of dual-task interventions to improve both single and dual-task gait speed [78]. The findings from the meta-analysis of Cinnera et al. support dual-task training as a beneficial therapy for improving dynamic balance and functional mobility in people with MS [79]. On the other hand, the impact of dual-task training in PD on dual-task gait speed is controversial [80,81]. The difference between performances in the two disease groups could be influenced by different brain substrates of neuroplasticity in the two diseases. It is well known that both PD and MS have the intrinsic property to structurally and functionally reorganize the damaged brain networks in response to rehabilitation [82,83]. The recovery in mild to moderate MS is achieved and sustained by the repair of damage through remyelination, with a resolution of inflammation and functional reorganization [84]. Whereas in PD the exercise, practice and movement strategy training act on supplementary circuits, thus supporting the sensorimotor integration and reinforcing the coupling of premotor areas, which are typically affected early in PD [85].

Our study had some limitations that are worth mentioning: it included patients with mild disability (H&Y score between 2.5 and 3 in people with PD; EDSS score ≤ 6.5 in people with MS) and the non-immersive virtual reality telerehabilitation system implemented in this study did not focus on upper limb motor therapy and specific cognitive rehabilitation. Therefore, the future research agenda should analyze the efficacy of telerehabilitation in more severely affected patients and increase the panel typology of treatments delivered remotely for complete physical and cognitive telerehabilitation in patients with chronic neurological diseases. Furthermore, the observed dropout rate (slightly more than 10%) was higher than the estimated *a priori* dropout rate (about 5%), approximately expected in these two neurological populations after virtual reality clinical trials as reported from recent metanalyses [86,87]. In these metanalyses, the typical reasons for dropout were difficulties in reaching the research center, refusal to participate, personal or familial issues, loss of data due to administrative problems, exacerbation of symptoms or other medical complications [86]. However, it is worth noting that in the period in which our study was carried out intervened a worldwide unpredictable adverse event due to the pandemic COVID-19. It has now been acknowledged that the COVID-19 pandemic has hindered the progress and completion of clinical trials [88].

## 5. Conclusions

The recent COVID-19 pandemic has underpinned the importance of ensuring a continuum of rehabilitation interventions, in particular for a frail population such as people with PD and MS. While therapeutic intervention in a clinical setting represents the first-line treatment, telerehabilitation intervention could act as the missing parts of the puzzle leading to an optimal continuity of care. Our results demonstrated that the non-immersive virtual reality telerehabilitation system is well tolerated and has positive effects on static and dynamic balance and gait in both people with PD and MS.

## Figures and Tables

**Figure 1 jcm-12-03178-f001:**
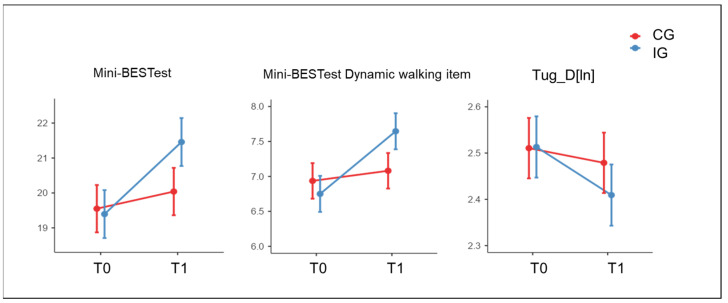
Statistically significant Time✻Group effects (*p*-value < 0.05) of the Intervention Group (IG) and of the Control Group (CG) in the total score of the mini-Balance Evaluation Systems Test (Mini-BESTest), Mini-BESTest Dynamic walking subcomponent, and Timed Up and Go Dual task (TUG-D) assessed before (T0) and after (T1) treatment. In TUG-D, the natural logarithm transformation [nl] was applied to take into account the absence of normal distribution. Data are represented as estimated marginal mean and standard error.

**Figure 2 jcm-12-03178-f002:**
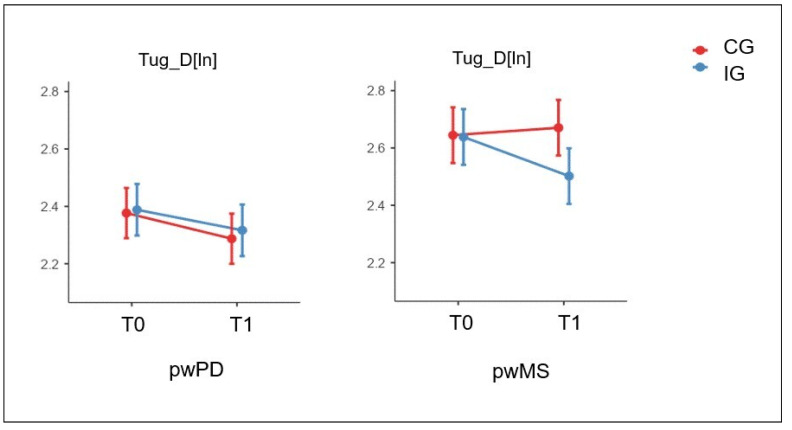
Statistically significant Time✻Group and Time✻Group✻Pathology effects (*p*-value < 0.05) of the Intervention Group (IG) and of the Control Group (CG) in the Timed Up and Go Dual task (TUG-D) assessed before (T0) and after (T1) treatment. Data from people with Parkinson’s Disease and Multiple Sclerosis are depicted separately. The natural logarithm transformation [nl] was applied to take into account the absence of normal distribution. Data are represented as estimated marginal mean and standard error.

**Table 1 jcm-12-03178-t001:** Summary of the subcomponents and the items of the mini-Balance Evaluation Systems Test (mini-BESTest).

Mini-BESTest (Max 28 Points)	
Anticipatory postural control (max 6 points) - Sit to stand - Rise to toes - Stand on one leg (right and left) *	Somatosensory orientation (max 6 points) - Stance on firm surface; eyes open - Stance on foam surface; eyes closed - Stance on incline surface, eyes closed
Reactive postural control (max 6 points) - Compensatory stepping correction; forward - Compensatory stepping correction; backward - Compensatory stepping correction; lateral (right and left) *	Dynamic walking (max 10 points) - Change in gait speed - Walk with head turns, horizontal plane - Walk with pivot turns - Step over obstacles - Timed up & go with dual task

* The worst side score was used to calculate the total score.

**Table 2 jcm-12-03178-t002:** Baseline demographics and clinical data in the full sample, people with Parkinson’s Disease and people with Multiple Sclerosis.

	Variables	IG	CG	Group Comparison [*p*-Value]
**Full sample (N = 132)**		**N = 65**	**N = 67**	
	Age years, [M, (SD)]	58.12 (12.43)	61.12 (11.06)	0.146 ^§^
	Education, N (%)			
	Primary	0 (0%)	3 (4.5%)	0.288 ^^^
	Secondary	13 (20.0%)	17 (25.4%)	
	High School	36 (55.4%)	33 (49.3%)	
	College	16 (24.6%)	14 (20.9%)	
	Sex (Male/Female), N%	29 (44.6%)/36 (55.4%)	30 (44.8%)/37 (55.2%)	0.985 ^^^
	Mini-BESTest, [M, (SD)]—primary outcome	19.43 (5.75)	19.70 (5.98)	0.791 ^§^
	Mini-BESTest Anticipatory postural control, [M, (SD)]	3.94 (1.55)	3.93 (1.54)	0.961 ^§^
	Mini-BESTest Reactive postural control, [M, (SD)]	4.22 (1.84)	4.21 (1.90)	0.984 ^§^
	Mini-BESTest Somatosensory orientation, [M, (SD)]	4.55 (1.48)	4.58 (1.43)	0.911 ^§^
	Mini-BESTest Dynamic walking, [M, (SD)]	6.74 (2.05)	6.99 (2.27)	0.514 ^§^
	TUG [ln], [M, (SD)]	2.29 (0.54)	2.33 (0.49)	0.668 ^§^
	TUG-D [ln], [M, (SD)]	2.50 (0.57)	2.50 (0.52)	0.942 ^§^
	MoCA, [M, (SD)]	25.88 (2.67)	25.19 (3.22)	0.341 *
**people with PD (N = 72)**		**N = 35**	**N = 37**	
	Age years, [M, (SD)]	66.51 (7.37)	68.32 (5.89)	0.252 ^§^
	Education, N (%)			
	Primary	0 (0%)	3 (8.1%)	0.166 ^^^
	Secondary	9 (25.7%)	12 (32.4%)	
	High School	16 (45.7%)	17 (45.9%)	
	College	10 (28.6%)	5 (13.5%)	
	Sex (Male/Female), N%	17 (48.6%)/18 (51.4%)	18 (48.6%)/19(51.4%)	0.995 ^^^
	H&Y, [median, (25th–75th)]	2.00 (2.00–2.00)	2.00 (1.50–2.50)	0.340 ^°^
	Disease Duration years [M, (SD)]	5.84 (4.57)	4.70 (3.76)	0.331 ^§^
	Mini-BESTest, [M, (SD)]—primary outcome	19.86 (5.60)	21.00 (5.52)	0.386 ^§^
	Mini-BESTest Anticipatory postural control, [M, (SD)]	4.20 (1.57)	4.05 (1.51)	0.689 ^§^
	Mini-BESTest Reactive postural control, [M, (SD)]	4.37 (1.63)	4.54 (1.64)	0.662 ^§^
	Mini-BESTest Somatosensory orientation, [M, (SD)]	4.71 (1.38)	5.00 (1.27)	0.364 ^§^
	Mini-BESTest Dynamic walking, [M, (SD)]	6.60 (1.99)	7.41 (1.95)	0.087 ^§^
	TUG [ln], [M, (SD)]	2.15 (0.54)	2.17 (0.50)	0.856 ^§^
	TUG-D [ln], [M, (SD)]	2.39 (0.58)	2.38 (0.55)	0.932 ^§^
	MoCA, [M, (SD)]	25.51 (2.66)	24.76 (3.09)	0.405 *
	MDS-UPDRS part III, [median, (25th–75th)]	27.00 (18.50–44.00)	33.00 (22.00–44.00)	0.388 ^°^
	PDQ-8, [M, (SD)]	28.75 (18.82)	27.96 (15.62)	0.845 ^§^
**people with MS (N = 60)**		**N = 30**	**N = 30**	
	Age years, [M, (SD)]	48.33 (9.66)	52.23 (9.34)	0.117 ^§^
	Education, N (%)			
	Primary	0 (0%)	0 (0%)	0.561 ^^^
	Secondary	4 (13.3%)	5 (16.7%)	
	High School	20 (66.7%)	16 (53.3%	
	College	6 (20.0%)	9 (30.0%)	
	Sex (Male/Female), N%	12 (40.0%)/18 (60.0%)	12 (40.0%)/18 (60.0%)	1.000 ^^^
	EDSS, [median, (25th–75th)]	5.00 (3.63–6.00)	4.50 (3.50–5.88)	0.634 ^°^
	MS Phenotype (RR/SP), N%	RR (13, 43%)/SP (17; 57%)	RR (16; 53%)/SP (14; 47%)	0.438 ^^^
	Disease Duration years [M, (SD)]	15.36 (7.17)	12.68 (6.72)	0.618 ^§^
	Mini-BESTest, [M, (SD)]—primary outcome	18.93 (5.98)	18.10 (6.23)	0.599 ^§^
	Mini-BESTest Anticipatory postural control, [M, (SD)]	3.63 (1.50)	3.77 (1.59)	0.739 ^§^
	Mini-BESTest Reactive postural control, [M, (SD)]	4.03 (2.08)	3.80 (2.12)	0.669 ^§^
	Mini-BESTest Somatosensory orientation, [M, (SD)]	4.37 (1.59)	4.07 (1.46)	0.449 ^§^
	Mini-BESTest Dynamic walking, [M, (SD)]	6.90 (2.14)	6.47 (2.56)	0.479 ^§^
	TUG [ln], [M, (SD)]	2.45 (0.50)	2.51 (0.41)	0.570 ^§^
	TUG-D [ln], [M, (SD)]	2.64 (0.55)	2.64 (0.44)	0.961 ^§^
	MoCA, [M, (SD)]	26.30 (2.67)	25.73 (3.34)	0.510 ^§^
	MSQOL-54_PHCS, [M, (SD)]	55.58 (16.47)	53.09 (19.13)	0.590 ^§^
	MSQOL-54_MHCS, [M, (SD)]	67.94 (16.59)	58.58 (22.10)	0.069 ^§^

Legend: PD = Parkinson’s Disease; MS = Multiple Sclerosis; CG = Control Group; IG = Intervention Group; M = mean; SD = standard deviation; Mini-BES Test = mini-Balance Evaluation Systems Test; TUG = Timed Up & Go; TUG-D = Timed Up-and-Go-test Dual-task; MoCA = Montreal Cognitive Assessment; MDS-UPDRS part III = MDS-Unified Parkinson’s Disease Rating Scale; PDQ-8 = Parkinson’s Disease Questionnaire; MSQOL-54 PHCS = Multiple Sclerosis Quality of Life-54 Physical Health Composite Score; MSQOL-54_ MHCS = Multiple Sclerosis Quality of Life-54 Mental Health Composite Score. RR = Relapsing-Remitting; SP = Secondary Progressive. Statistical comparisons between IG and CG were performed using Independent sample *t*-test (^§^); ANCOVA adjusting for age and education (*); Mann-Whitney U test (^°^); Chi-square (^^^); ln = Logarithm Natural transformation was applied to account for no normal distribution. Statistically significant results (*p*-value < 0.05).

**Table 3 jcm-12-03178-t003:** Results of the Generalized Linear Mixed Models in the Intervention Group (IG) and in the Control Group (CG).

Variables	IG (N = 65)				CG (N = 67)				Time [*p*-Value]	Group [*p*-Value]	Time✻Group [*p*-Value]
	EMM T0	SE T0	EMM T1	SE T1	EMM T0	SE T0	EMM T1	SE T1			
Mini-BESTest—primary outcome	19.40	0.69	21.46	0.69	19.55	0.68	20.04	0.68	**<0.001**	0.487	**0.020**
Mini-BESTest Anticipatory postural control	3.92	0.18	4.51	0.18	3.91	0.18	4.12	0.18	**<0.001**	0.394	0.082
Mini-BESTest Reactive postural control	4.20	0.22	4.46	0.22	4.17	0.22	4.26	0.22	0.207	0.688	0.554
Mini-BESTest Somatosensory orientation	4.54	0.17	4.85	0.17	4.53	0.16	4.57	0.16	0.055	0.513	0.137
Mini-BESTest Dynamic walking	6.75	0.26	7.65	0.26	6.94	0.25	7.08	0.25	**<0.001**	0.568	**0.011**
TUG [ln]	2.30	0.06	2.23	0.06	2.34	0.06	2.31	0.06	**0.002**	0.469	0.250
TUG-D [ln]	2.51	0.07	2.41	0.07	2.51	0.07	2.48	0.07	**<0.001**	0.714	**0.048**
MoCA	25.85	0.35	26.62	0.35	25.29	0.34	25.85	0.34	**0.003**	0.125	0.616

Legend: PD = Parkinson’s Disease; MS = Multiple Sclerosis; IG = Intervention Group; CG = Control Group; EMM = Estimated Marginal Mean; SE = Standard Error; T0 = baseline; T1 = end of treatment; Mini-BESTest = mini-Balance Evaluation Systems Test; TUG = Timed Up and Go; TUG-D = Timed Up and Go Dual task; MoCA = Montreal Cognitive Assessment; ln = Logarithm Natural transformation was applied to account for no normal distribution; Statistical significant results (*p*-value < 0.05) are highlighted in bold font.

## Data Availability

The data associated with this paper are not publicly available but are available from the corresponding author upon reasonable request.

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
