# Peer review of "Non-Immersive Virtual Reality Telerehabilitation System Improves Postural Balance in People with Chronic Neurological Diseases"

_jcm, 2023, doi:10.3390/jcm12093178_

Round 1

Reviewer 1 Report

The manuscript describes the effects of a TeleRehabilitation intervention on various motorskil and cognitive outcomes of patients with multiple sclerosis and with Parkinson's disease. The topic is of interest and of clinical value, as these patients have significant problems with mobility and finding a way to deliver efficacious exercise interventions remotely would have a significant clinical impact. However, the manuscript has to be improved in the following areas:

1) the Introduction and the Discussion are written in a single paragraph and they are very difficult to follow in their current format. The authors need to break down their messages into paragraphs of congruent conceptual meaning. Currently the readers is not sure where one argument ends and the next begins.

2) The use of acronyms is excessive, and at times annoying, and should be reduced to the minimum. The authors are not saving much word count, instead the reader has to juggle in their memory all the different acronyms or go back and forth to find out where each acronym was introduced as a full term the first time.

3) The inclusion and exclusion criteria should be better organized. Currently, the reader has to go back and forth between inclusion and exclusion criteria and for either disease.

4) The authors state that the exercise regimen was tailored to patients characteristics and needs. Can the authors please provide which characteristics and needs were chosen for tailoring purposes?

5) The statistical analysis was set to detect an effect size of 1.0 (Cohen's d=1.0). This is a large effect size, given that the control group also received tailored exercises. Can the authors please explain how they decided on such an expected effect size? Based on literature, on pilot data, etc.?

6) The statistical methods mention that missing data were imputed with the mean value of the given variable, but the % of missing data has not been reported, unless this reviewer missed it.

7) Along these lines, it is not clear why the authors included both diseases in their analysis, as if they have already analyzed outcomes for the multiple sclerosis group (reference 38)

Author Response

Reviewer 1

The manuscript describes the effects of a TeleRehabilitation intervention on various motor skills and cognitive outcomes of patients with multiple sclerosis and with Parkinson's disease. The topic is of interest and of clinical value, as these patients have significant problems with mobility and finding a way to deliver efficacious exercise interventions remotely would have a significant clinical impact. However, the manuscript has to be improved in the following areas:

Thank you for the revision and suggestions.

1) the Introduction and the Discussion are written in a single paragraph and they are very difficult to follow in their current format. The authors need to break down their messages into paragraphs of congruent conceptual meaning. Currently the readers is not sure where one argument ends and the next begins.

R1: Thanks for the suggestion. We revised the manuscript accordingly.

2) The use of acronyms is excessive, and at times annoying, and should be reduced to the minimum. The authors are not saving much word count, instead the reader has to juggle in their memory all the different acronyms or go back and forth to find out where each acronym was introduced as a full term the first time.

R2: Thank you for the revision. We agree with you, and we reduced the acronyms in the text.

3) The inclusion and exclusion criteria should be better organized. Currently, the reader has to go back and forth between inclusion and exclusion criteria and for either disease.

R3: We agree with you, and we re-organized the inclusion/exclusion criteria.

4) The authors state that the exercise regimen was tailored to patients characteristics and needs. Can the authors please provide which characteristics and needs were chosen for tailoring purposes?

R4: The intensity and duration of the CG were the same as the TG. Both groups conducted 45 minutes of exercises for each training session and performed the same type of exercises: the typologies of exercises (e.g., maintaining balance on one leg, marching in place, standing on tiptoe, squatting, etc…) were the same but the duration of every single exercise was personalized based on the patient's attention, compliance, and balance impairment level so that he/she did not get tired or lost concentration.

5) The statistical analysis was set to detect an effect size of 1.0 (Cohen's d=1.0). This is a large effect size, given that the control group also received tailored exercises. Can the authors please explain how they decided on such an expected effect size? Based on literature, on pilot data, etc.?

R5: This secondary analysis was carried out on a sample of people with MS included in the previously study (see Pagliari et al., 2021) and a sample of PD group in add-on. For the subsample of people with MS the calculation of the a priori sample size was reported in Pagliari et al., 2021. For the subsample of people with PD, we estimated the a priori sample size on the primary outcome MiniBES Test according to previously published results reporting a large effect size (Cohen’s d = 1) after a balance training in people with PD (Leavy et al., 2020 10.1097/NPT.0000000000000298). In a design with two independent groups (Experimental vs Control), a sample size of 60 patients (30 per arm) achieves 95% power to detect a difference of 2.0 in the experimental group after treatment (standard deviation= 2.0; Cohen’s d= 1), assuming an alpha error of 0.05, and 5% dropout rate of patients. In Leavy et al. (2020), the control group received no clinical structured rehabilitation intervention, as in our trial, for this reason we expected a large effect size. A priori sample size was calculated using G*Power 3.1 software (Faul et al., 2007 DOI: 10.3758/bf03193146). We better specified this issue in the paragraph 2.4. Statistical analysis.

6) The statistical methods mention that missing data were imputed with the mean value of the given variable, but the % of missing data has not been reported, unless this reviewer missed it.

R6: According to reviewer, we calculated the proportion of data missing of the whole sample (2.75%). Considering that this proportion of missing data is less than of 5%, we performed a single imputation as reported by Jakobsen et al., 2017 (DOI: 10.1186/s12874-017-0442-1).

7) Along these lines, it is not clear why the authors included both diseases in their analysis, as if they have already analyzed outcomes for the multiple sclerosis group (reference 38)

R7: In this secondary analysis, we considered the Tug and TugDT, which were not reported in the previously published trial (Pagliari et al., 2021). Furthermore, we tested the effect of treatment on two different neurological diseases which share some characteristics: are both highly disabling conditions, involve motor and non-motor symptoms, which culminate in balance and gait impairments and are accompanied with reduced quality of life. The inclusion of participants affected by these two different clinical conditions allows us to test not only the general impact of treatment and time (main effects of Group and Time), but also to check for potential interactions between the effect of the treatment and the type of pathology (interaction effect Group by Pathology by Time) on the selected outcomes. The magnitude of the beneficial impact of our VR experimental intervention might be higher in one of the two considered neurological diseases considering some intrinsic differences between them (aetiopathogenesis, mean age of patients, …).

Reviewer 2 Report

Goffredo et al. evaluated the efficacy of a non-immersive virtual reality TR system on postural balance in people with PD and MS, compared to at-home conventional rehabilitation. The rationale of this study is extremely relevant. The study is well designed and the article is clearly written and organized. 

However, some concerns:

-        The authors enrolled patients with both RRMS and SPMS. Despite, the randomization process not is clear whether the was a difference in phenotype distribution between the intervention and control group. If there were more progressive patients in the control group, this might have introduced a bias responsible for the greater improvement observed in subjects in the intervention group. In line, are there differences in terms of disease duration between groups? Did you account for in the linear mixed model?

-        The authors presented in table the mean age and mean EDSS; however, they should present also phenotype distribution and disease duration.

-        EDSS and UPDRS reported in the table: The EDSS and IPDRS shown in the table correspond to what time point? Why were these measures not used as secondary outcomes?

-        “A dataset including 132 participants, who completed the baseline (T0) and post-treatment (T1) evaluations, was considered in these analyses (72 pwPD; 60 pwMS).” How many patients did not complete the treatment?

Minor concerns:

-        I would suggest shortening the introduction section, but at the same time to introduce better the role of digital technology in the treatment of chronic neurological disorders.

-        Line 224: “and the following ore for”; amend it.

Author Response

Reviewer 2

Comments and Suggestions for Authors

Goffredo et al. evaluated the efficacy of a non-immersive virtual reality TR system on postural balance in people with PD and MS, compared to at-home conventional rehabilitation. The rationale of this study is extremely relevant. The study is well designed and the article is clearly written and organized. 

Thank you for the revision and suggestions.

However, some concerns:

1) The authors enrolled patients with both RRMS and SPMS. Despite, the randomization process not is clear whether the was a difference in phenotype distribution between the intervention and control group. If there were more progressive patients in the control group, this might have introduced a bias responsible for the greater improvement observed in subjects in the intervention group. In line, are there differences in terms of disease duration between groups? Did you account for in the linear mixed model?

R1: In response to the reviewer’s requests, the distribution of RRMS and SPMS phenotype across the two groups was calculated. No statistically significant differences were found between the experimental and control groups. Similarly, no differences for disease duration were found in the MS and PD groups between the experimental and control groups. For these reasons, these variables were not included in the linear mixed model as covariates.

2) The authors presented in table the mean age and mean EDSS; however, they should present also phenotype distribution and disease duration.

R2: In accordance with the reviewer’s request, we included the multiple sclerosis phenotype and the disease duration in Table 2.

3)  EDSS and UPDRS reported in the table: The EDSS and UPDRS shown in the table correspond to what time point? Why were these measures not used as secondary outcomes?

R3: The EDSS and UPDRS scores in Table 2 were from baseline (T0), as described in the table caption. EDSS and UPDRS scales were considered as eligibility criteria.

4)  “A dataset including 132 participants, who completed the baseline (T0) and post-treatment (T1) evaluations, was considered in these analyses (72 pwPD; 60 pwMS).” How many patients did not complete the treatment?

R4: Thank you for the comment. We described the participants recruitment and the drop-out rate more in details in the text as follows:

A sample of 150 participants met the inclusion criteria and were included in the study: 80 people with PD and 70 with SM. Of the sample, 75 participants were allocated to the TG (PD=40; MS=35) and 75 to the CG (PD=40; MS=35). A sample of 132 participants (TG=65; CG=67) completed the baseline (T0) and post-treatment (T1) evaluations, and 18 subjects dropped out (TG=10; CG=8). None of the drop-outs occurred during the study for treatment-related reasons, and no participant experienced any adverse event during treatment.

5) I would suggest shortening the introduction section, but at the same time to introduce better the role of digital technology in the treatment of chronic neurological disorders.

R5: Thank you for your suggestion. We have revised the Introduction in accordance with your comment.

6) Line 224: “and the following ore for”; amend it.

R6: Thank you. We corrected it.

Reviewer 3 Report

In their paper Goffredo et al. investigated the efficacy of a non-immersive virtual reality TeleRehabilitation system on postural balance in people with PD and MS, compared to at-home conventional rehabilitation. In complex, the study is well-designed and clearly presented. However, minor changes are suggested.

Introduction: well-organized and well-written

1)      Some acronyms do not have a previous specification (WHO, RCT).

2)      Line 76: impose.

Materials and Methods: informative and complete

3)      Line 214-219: describing MDS-UPDRS, an explanation of part IV is omitted. Has this subscale also been administered?

4)      Line 241-245: the paragraph lacks a verb.

Results: clearly exposed and well-analysed

5)      Line 282-283: it is reported “an improvement of about 22% with respect to the maximum score achievable” in the mini-BEST total score of the TG but the mean score increases of 2.06 points out of 28 which is around 7%.

6)      Line 293-294: similarly, “an improvement of about 25% with respect to the maximum score achievable in both groups” in the MOCA score is reported, but in the TG the mean score increases of 0.77 points out of 30 which is around 2.6% and in the CG the mean score increases of 0.56 points out of 30 which is around 1.9%. It's correct?

7)      In table 2: regarding the test used to assess a possible difference between groups in the sex variable, it is reported that independent sample t-test was used but, given the categorial variable, why didn't you use the chi-square test?

8)      In table 2: reporting sex variable of the CG in the pwPD brackets are misplaced.

9)      In table 2: I would have added median score of part I, II and III of MDR-UPDRS. Have this subscales score been analysed in order to assess the absence of any significant difference in the two groups?

10)   In table 2: the reported MOCA mean of the TG in the pwMS is 16.30. Given that the cut-off score chose was 18 and the lack of a significant difference with the CG group, I suppose that it is a typo and you meant 26.30.

11)   Line 298-304: as shown in figure 2, I suppose that no difference in the improvement in the TUG-D score after treatment between the two groups was observed also in pwPD but this analysis is specified only for pwMS.

12)   Have you investigated if there was any significant difference regarding demographic variables as well as motor and cognitive status between pwPD and pwMS at baseline? I think that an analysis of this aspect would further confirm the rationale under the decision of analysing together the two groups of CNDs patients (as supported also in line 378-383 of the discussion).

Author Response

Reviewer 3

In their paper Goffredo et al. investigated the efficacy of a non-immersive virtual reality TeleRehabilitation system on postural balance in people with PD and MS, compared to at-home conventional rehabilitation. In complex, the study is well-designed and clearly presented. However, minor changes are suggested.

Introduction: well-organized and well-written

1) Some acronyms do not have a previous specification (WHO, RCT).

R1: In accordance with the reviewer’s request we specified acronyms.

2) Line 76: impose.

R2: Thank you. We corrected it.

Materials and Methods: informative and complete

3) Line 214-219: describing MDS-UPDRS, an explanation of part IV is omitted. Has this subscale also been administered?

R3: In accordance with the reviewer’s request we insert the explanation of part IV.

4) Line 241-245: the paragraph lacks a verb.

R4: Thank you. We corrected it.

Results: clearly exposed and well-analysed

5) Line 282-283: it is reported “an improvement of about 22% with respect to the maximum score achievable” in the mini-BEST total score of the TG but the mean score increases of 2.06 points out of 28 which is around 7%.

R5: Thank you for the comment. As suggested by the reviewer, in the revised version of the manuscript we adopted the estimated marginal means (EMM) reported in Table 3 to perform the calculation, instead of descriptive means. For the calculation of the percentage of improvement in the TG group we used the following proportion: (Maximum change achievable: 100% = Observed change: x%). In more detail, the maximum change achievable was obtained by subtracting the EMM at baseline (19.40) from the maximum score at the mini-BEST total score (28) (Maximum score of mini-BEST - EMM mini-BEST T0: 28 – 19.40 = 8.60). Then, in order to calculate the x%, the observed change in the TG group (2.06) was divided by the maximum change achievable (8.60). The result indicates an improvement of about 24% with respect to the maximum score achievable.

6) Line 293-294: similarly, “an improvement of about 25% with respect to the maximum score achievable in both groups” in the MOCA score is reported, but in the TG the mean score increases of 0.77 points out of 30 which is around 2.6% and in the CG the mean score increases of 0.56 points out of 30 which is around 1.9%. It's correct?

R6: As detailed in the 5) response, for the calculation of the percentage of improvement in the TG group we used the following proportion: (Maximum change achievable: 100% = Observed change: x%). In more detail, the maximum change achievable was obtained by subtracting the EMM at baseline (25.85) from the maximum score at the MoCA total score (30) (Maximum score of MoCA - EMM MoCA T0: 30 – 25.85 = 4.15). Then, in order to calculate the x%, the observed change in the TG group (0.77) was divided by the maximum change achievable (4.15). The result indicates an improvement of about 19% with respect to the maximum score achievable.

7) In table 2: regarding the test used to assess a possible difference between groups in the sex variable, it is reported that independent sample t-test was used but, given the categorial variable, why didn't you use the chi-square test?

R7: Thank you for the comment, it was the typo error. We corrected it.

8) In table 2: reporting sex variable of the CG in the pwPD brackets are misplaced.

R8: Thank you for the comment. We corrected it.

9) In table 2: I would have added median score of part I, II and III of MDR-UPDRS. Have this subscales score been analysed in order to assess the absence of any significant difference in the two groups?

R9: Thank you for the comment. We used the MDR-UPDRS part III, we corrected it in the table 1.

10) In table 2: the reported MOCA mean of the TG in the pwMS is 16.30. Given that the cut-off score chose was 18 and the lack of a significant difference with the CG group, I suppose that it is a typo and you meant 26.30.

R10: Thank you, it was a typo error. We corrected it.

11) Line 298-304: as shown in figure 2, I suppose that no difference in the improvement in the TUG-D score after treatment between the two groups was observed also in pwPD but this analysis is specified only for pwMS.

R11: Thank you. We specified it.

12) Have you investigated if there was any significant difference regarding demographic variables as well as motor and cognitive status between pwPD and pwMS at baseline? I think that an analysis of this aspect would further confirm the rationale under the decision of analysing together the two groups of CNDs patients (as supported also in line 378-383 of the discussion).

R12: Thank you for your suggestion, we calculated the difference regarding demographic variables and primary outcome between pwPD and pwMS at baseline. We added the Table S1 with the results in the supplementary materials.

Round 2

Reviewer 1 Report

Thank you for revising the manuscript to improve readability.

Regarding the use of acronyms, the authors do not need to define WHO and RCT - these are well known acronyms in health sciences. However, in some places they use CNDs and in other they use "chronic neurological disorders". Please, consistently write "chronic neurological disorders" and avoid creating a new acronym. Similarly, they refer to TR for the intervention (Telerehabilitation) and TG for the intervention group (Telerehabilitation Group). Please, always mention telerehabilitation and use IG and CG for the intervention and control groups, respectively.

The Inclusion and Exclusion criteria are still written as a "soup", with repetitions and omissions. Maybe it will help the authors to organize this section if they describe the process of selecting participants

a.) either chronologically (e.g., did they sign the consent form before eligibility assessments) or

b.) based on what information they collected from medical files (e.g., diagnosis of Parkinson's disease, stable drug treatment etc.) and what from testing participants themselves

Regarding the criterion: sufficient cognitive level to understand the study procedures - was this assessed with MoCA? or how else?

The absence of signature in the consent form should not be listed as an exclusion criterion.

Regarding the power of the study - the authors mention that their power calculation assumed a drop out rate of 5%. However, the drop out rate was more than 10% (lines 390-396).  Was the study under powered to detect intervention outcomes? This was not mentioned in the limitations.

Minor points:

Line 69: revise "his care plan" to either "his or her care plan" or to "their care plan"

Line 329: the sentence is strange, starting with "and the following ones..."

Author Response

Thank you for revising the manuscript to improve readability.

1) Regarding the use of acronyms, the authors do not need to define WHO and RCT - these are well known acronyms in health sciences. However, in some places they use CNDs and in other they use "chronic neurological disorders". Please, consistently write "chronic neurological disorders" and avoid creating a new acronym. Similarly, they refer to TR for the intervention (Telerehabilitation) and TG for the intervention group (Telerehabilitation Group). Please, always mention telerehabilitation and use IG and CG for the intervention and control groups, respectively.

R1: Thank you for the suggestions, we modified it.

2) The Inclusion and Exclusion criteria are still written as a "soup", with repetitions and omissions. Maybe it will help the authors to organize this section if they describe the process of selecting participants

a.) either chronologically (e.g., did they sign the consent form before eligibility assessments) or  b.) based on what information they collected from medical files (e.g., diagnosis of Parkinson's disease, stable drug treatment etc.) and what from testing participants themselves

R2: Thank you the reviewer, we modified the text as suggested.

3) Regarding the criterion: sufficient cognitive level to understand the study procedures - was this assessed with MoCA? or how else?

R3: The sufficient cognitive level was evaluated by MoCA Test. According to the Italian normative value, a total score ≥18 was considered a normal global cognitive level score (see ref Santangelo et al 2015)

4) The absence of signature in the consent form should not be listed as an exclusion criterion.

R4: Thank you for the revision, we modified it.

5) Regarding the power of the study - the authors mention that their power calculation assumed a drop out rate of 5%. However, the drop out rate was more than 10% (lines 390-396).  Was the study under powered to detect intervention outcomes? This was not mentioned in the limitations.

R5: The observed dropout rate (slightly more than 10%) was higher than the estimated a priori dropout rate (about 5%), approximately expected in these two neurological populations after virtual reality clinical trials as reported from recent metanalyses (Casuso-Holgado et al., 2022, https://doi.org/10.1007/s10055-022-00733-4; Parra et al., 2023, https://doi.org/10.1007/s10916-023-01930-7). In these works, the typical reasons for dropout were difficulties in reaching the research centre, refusal to participate, personal or familial issues, loss of data due to administrative problems, exacerbation of symptoms or other medical complications (Casuso-Holgado et al., 2022). However, it is worth noting that in the period in which our study was carried out intervened a worldwide unpredictable adverse event due to the pandemic COVID-19. It has now been acknowledged that the COVID-19 pandemic has hindered the progress and completion of clinical trials (Chen et al., 2021, https://doi.org/10.1371/journal.pone.0251410). We added this paragraph in the limitation section.

Minor points:

6)Line 69: revise "his care plan" to either "his or her care plan" or to "their care plan"

R6: Thank you for the revision, we modified it.

7)Line 329: the sentence is strange, starting with "and the following ones..."

R7: Thank you for the revision, we modified it.

Reviewer 2 Report

none

Author Response

Thank you